# The Isocyanurate-Carbamate-Bridged Hybrid Mesoporous Organosilica: An Exceptional Anchor for Pd Nanoparticles and a Unique Catalyst for Nitroaromatics Reduction

Ali Zebardasti [1], Mohammad G. Dekamin [1,*] and Esmail Doustkhah [2,*]

1   Pharmaceutical and Heterocyclic Compounds Research Laboratory, Department of Chemistry, Iran University of Science and Technology, Tehran 1684613114, Iran; zebardasti@nano.ir
2   International Center for Materials Nanoarchitechtonics (MANA), National Institute for Materials Science (NIMS), 1-1 Namiki, Tsukuba, Ibaraki 305-0044, Japan
*   Correspondence: mdekamin@iust.ac.ir (M.G.D.); Esmaildostkhah@gmail.com (E.D.)

**Abstract:** Hybridisation of mesoporous organosilicas (MO) to reinforce the surface capability in adsorption and stabilisation of noble metal nanoparticles is of great attention in generating/supporting noble metal within their matrices and transforming them into efficient heterogeneous catalysts. Here, we used a unique hybrid of organic-inorganic mesoporous silica in which pore profile pattern was similar to the well-known mesoporous silica, SBA-15 for catalysis. This hybrid mesoporous organosilica was further engaged as a support in the synthesis and stabilisation of Pd nanoparticles on its surface, and then, the obtained Pd-supported MO was employed as a heterogeneous green catalyst in the conversion of aqueous *p*-nitrophenol (PNP) to *p*-aminophenol (PAP) at room temperature with efficient recyclability.

**Keywords:** hybrid mesoporous silica; Pd nanoparticles; *p*-nitrophenol; reduction; heterogeneous catalyst; PMO

## 1. Introduction

Nanoporous silicas modified/constructed by organosiloxanes (MO) [1–5] are a class of hybrid silica-based materials that are widely used in the diverse variety of applications including catalysis [6,7], microextraction [8], water treatment [9], molecular recognition [10], photocatalysis [11], optical thermometry [12], sensors [13,14], and $CO_2$ capture [15,16]. These materials can be synthesised by condensation of various organosiloxane precursors and usually have amorphous structures but with ordered/disordered pore channels. Some of these materials can have microporosity alongside the mesoporosity, which can increase the surface area and subsequently the efficiency of the material in the applications [17,18]. Furthermore, in some cases, these materials are promising to metal-organic-frameworks in viewpoint of porosity and stability [5,19]. Since these materials are synthetic, there should be a synthetic bottom-up strategy for reaching to such materials in which the presence of template (mostly a soft template), e.g., F127 and P123, is inevitable [4,20,21].

When an organosiloxane bridge with significant flexibility is being used in a MO's structure, the organosiloxane bridge should co-condense with a silica source (e.g., tetraethyl orthosilicate (TEOS)) to generate a uniform and robust mesoporous structure from viewpoint of mechanical toughness and porosity [4]. Several advantages belong to MOs, which cannot be found in conventional hybrid mesoporous silica materials [22]. For instance, in MOs, depending on the type of bridge, a higher ratio of organosilica can be embedded in the structure of MO, while in the case of conventional mesopores, e.g., SBA-15, it is an overwhelming process to functionalise with organosiloxane precursors in high ratio that usually deals with pore blocking or unsuccessful process [23]. In MOs, the mass transfer in pores for guest molecules is more efficient and also more promising than conventional

hybrid silica mesopores [12,24,25]. This efficiency in transfer of molecules improves the life-time of the MO and catalytic activity through minimizing the pore-blocking possibility by guest molecules [26–28]. This is because the organosilica motif of conventional mesoporous silicas stay on the external surface of pore channels, while in MOs, it can be embedded in the pore wall [29].

The use of mesoporous silica materials is of great importance among the candidates for synthesis of heterogeneous catalysts [30–34]. These materials can provide excellent heterogeneous surface for immobilisation of catalytically active metal species for various reactions, such as cross coupling and reduction reactions [35–39]. These materials can also provide an excelling recyclability by tuning the ratio and type of the organosilica. Pd, among the noble metals, has played a pivotal role in the catalysis since Pd is an active catalytic species for a broad domain of reactions, e.g., cross-coupling [40,41], oxidation [42], reduction [6], and dehydrogenation [17,43]. Here, we integrated the isocyanurate and carbamate functional groups in the MO and used it for supporting of Pd nanoparticles. This was further employed as a heterogeneous green catalyst for the aqueous room temperature reduction of PNP to PAP.

## 2. Results and Discussion

### 2.1. Synthesis and the Characterisation of the Catalyst

We here synthesised a silica porous material by co-condensation method, where the organic and inorganic are homogeneously mixed and dissolved into a new material. Since this hybrid mesoporous silica material (MO-ISO) was obtained by co-condensation of a synthetic organosiloxane, we developed a facile one-step solvent-free approach to synthesise the organosiloxane precursor in our laboratory, as represented in Figure 1, and we further used it for immobilisation of Pd nanoparticles. These Pd nanoparticles were primarily obtained by adsorption of $Pd^{2+}$ ions on MO-ISO and eventual reduction of them with $NaBH_4$ at room temperature.

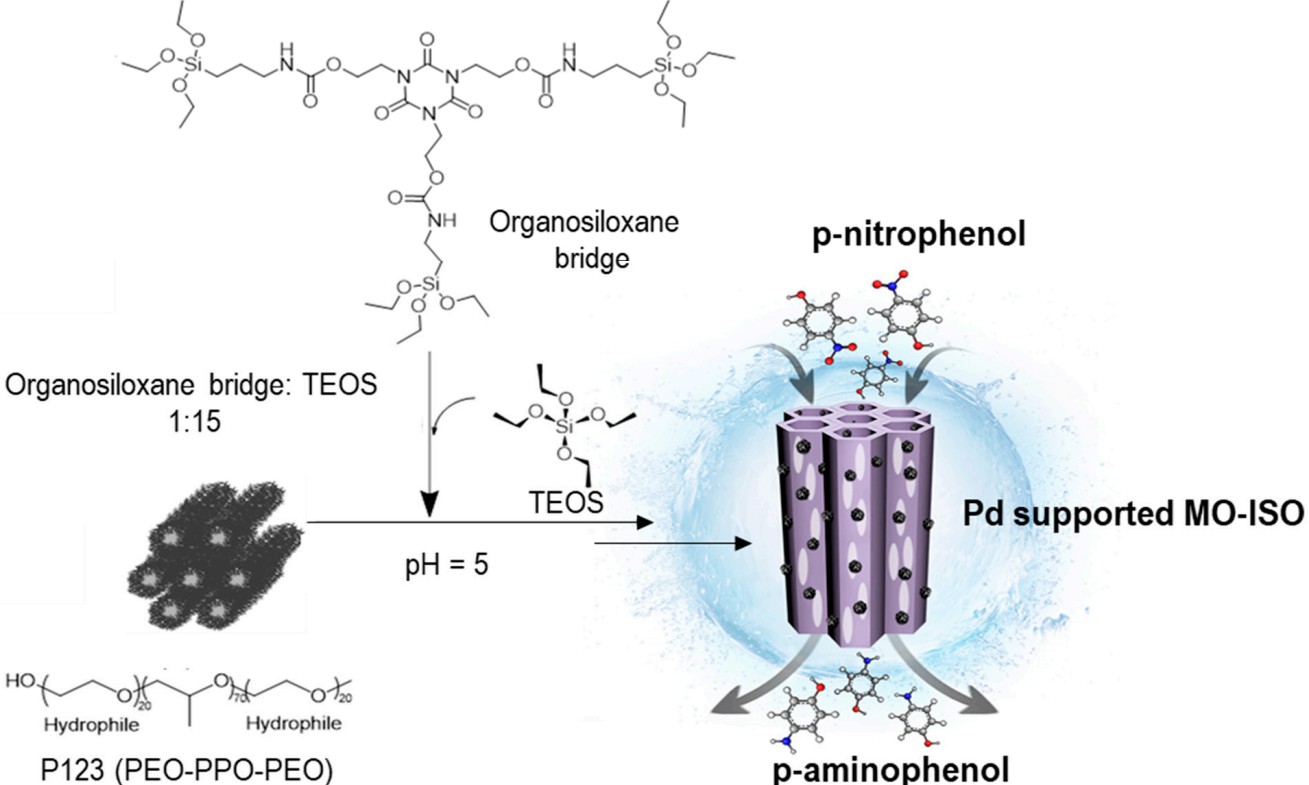

**Figure 1.** Schematic representation of Pd@MO-ISO synthesis.

We previously proved the successful synthesis of this organosiloxane bridge by various techniques such as $^1$H- and $^{13}$C-NMR, FTIR, and mass spectroscopy, which all have been discussed in the literature [16]. We used this isocyanurate-carbamate organosiloxane bridge (ISO bridge) to synthesise the mesoporous organosilica (MO-ISO) with a high surface area and mechanically stable properties. Since in the previous study, we showed that the ratio of TEOS to organosiloxane precursor has a critical effect on the surface area and morphology, we selected the molar ratio of 1:15 (organosiloxane to TEOS, respectively). For studying the surface area, the $N_2$ adsorption–desorption isotherms of the synthesised MO-ISO were evaluated and represented in Figure 2. Accordingly, the surface area is obtained around 697 m$^2 \cdot$g$^{-1}$ with type IV isotherm and the average pore size is 6.2 nm. Going further, the Pd-supported MO-ISO (Pd@MO-ISO) has exhibited a relative loss in the surface area (389 m$^2 \cdot$g$^{-1}$), however, the surface area is still high, compared to other porous materials (Figure 2). This decrease in the surface area can be attributed to the relative occupancy of the pore space through the Pd nanoparticles.

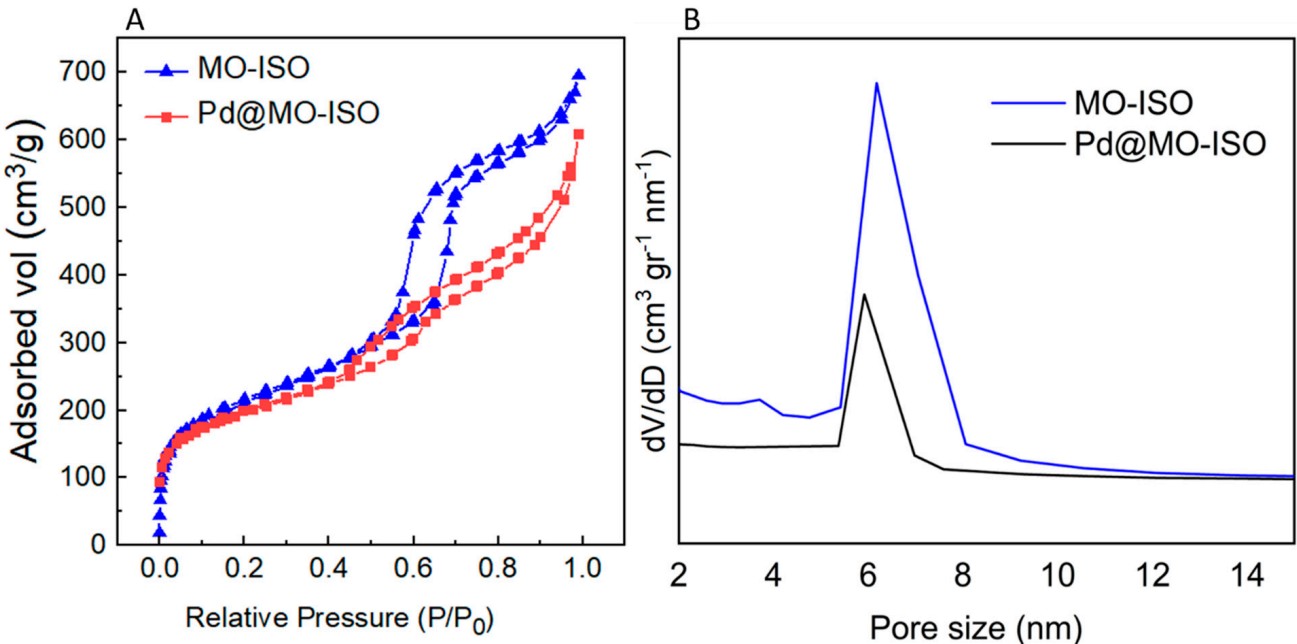

**Figure 2.** (**A**) $N_2$ adsorption–desorption isotherms and (**B**) BJH plots of MO-ISO and Pd@MO-ISO.

The FTIR spectrum for 1,3,5-tris(2-hydroxyethyl)-1,3,5-triazinane-2,4,6-trione (THEIC), the heterocyclic starting material of ISO bridge, was compared with the FTIR spectrum of the MO-ISO to see if the main structure of ISO has undergone any change (Figure 3). This can be judged by the existence of two sharp bands at 1467 and 1700 cm$^{-1}$ related to the stretching vibrations of the isocyanurate carbonyl located in the ring. A small shift in this regard can be observed to these band positions, which can be attributed to the change in the intermolecular hydrogen bondings in the pure form and when embedded in the MO structure. Some peaks in the range of 2900–3000 cm$^{-1}$ are also related to the aliphatic chains (ethylene) of the ISO bridge.

Since the generated carbamate groups are sensitive to the acidic and basic media, we carefully examined the appeared two peaks in the $^{13}$C-NMR spectrum of MO-ISO related to the carbonyls (one to carbamate (156 ppm) and another to the isocyanurate (148 ppm)), which can confirm that the ISO bridge has been retained intact in the structure [16].

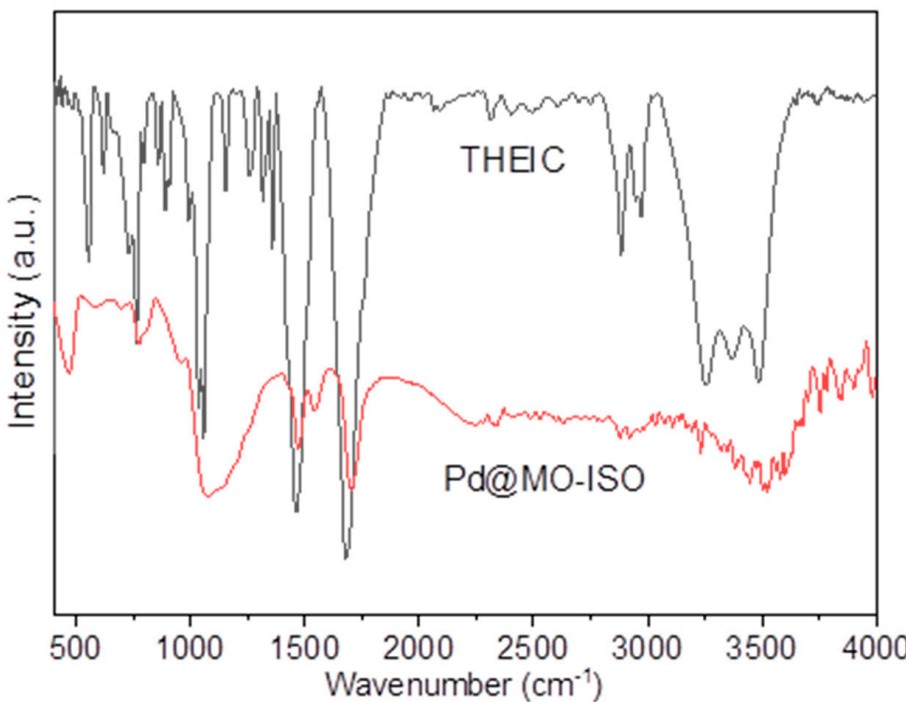

**Figure 3.** ATR-FTIR spectra of THEIC and Pd@MO-ISO.

SEM micrographs of MO-ISO morphology exhibits micro-sized particles, showing that the mesopores are assembled and aggregated into the large particles (Figure 4A). Furthermore, high-angle annular dark-field scanning transmission electron microscopy (HAADF-STEM) image of MO-ISO confirms the presence of Pd nanoparticles distributed in the MO-ISO's matrix. Note that the Pd nanoparticles are with lighter colour, since the more the condense matter, the lighter the appearance it takes in HAADF-STEM image (Figure 4B). Looking at the TEM image of MO-ISO reveals that the structure of MO-ISO is hexagonally ordered with uniform pore size distribution (Figure 4C). Moreover, the Pd-supported MO-ISO confirms the presence of Pd nanoparticles formed on the MO-ISO (Figure 4D). The calculated sizes of Pd nanoparticles according to the TEM and HAADF-STEM images are found to be ~12.9 nm (See inset Figure 4D).

In addition, we studied the Pd nanoparticles and the MO-ISO structure through elemental TEM-mapping and TEM-based electron-dispersive spectroscopy (EDS). We confirmed the supporting of Pd nanoparticles in MO-ISO structure by scanning the related elements such as O, N, C, and Si through TEM-mapping. This is illustrated in Figure 5A. Figure 5B also shows the TEM-EDS spectra of Pd@MO-ISO. This analysis confirms the presence of Pd element in the structure as well as other critical elements, e.g., C, N, and Si.

### 2.2. Catalytic Test

Further, we tested the catalytic activity of Pd@MO-ISO in the reduction of nitroaromatics by testing PNP in the aqueous media at room temperature. In this regard, we used sodium borohydride ($NaBH_4$) as reducing agent and $H_2O$ as a green solvent. The catalyst, Pd@MO-ISO, was used in different amounts to see which ratio of Pd to the reactant, PNP, has the higher TON. Accordingly, the results indicated that 5 mg catalyst in 3 mM PNP solution (40 mL) has a higher activity in terms of Turn Over Number (TON). TON was obtained by following formula:

$$\text{TON} = \frac{\text{generated product (mmol)}}{\text{active sites of the catalyst (mmol)}}, \tag{1}$$

where the TON value was obtained as 125 for the PNP reduction through 5 mg Pd@MO-ISO (with 2 wt% of Pd content) and 0.12 mmol PAP. Note that we assigned the active sites of the catalyst to entire number of Pd atoms in the structure, regardless of which the Pd atoms are the surficial or interior ones. Therefore, we neglected the small fraction of Pd atoms that are catalytically inactivated by outermost layers. Further, examinations indicated that the higher amount of the catalyst (10 mg) does not have significant effect on the improvement of the catalytic activity from viewpoint of TON. It is noteworthy that the lower loadings and higher loading of Pd (=0.5 wt% and 3 wt%) on MO-ISO did not have promising catalytic results since the TON of the reaction were obtained lower than that of 2 wt%-loading Pd nanoparticles. This observation indicates that the TON value is optimal at 2 wt% of loaded Pd in MO-ISO for the reduction of *p*-nitrophenol. In addition, we examined the reduction reaction within 20 min while sampling in every 5 min. We realised that the major fraction of the reaction progress occurs at early 10 min. We confirmed the conversion of PNP to PAP by observing fading of an adsorption peak at 410 nm and appearance of a related peak at 317 nm over the reaction time (Figure 6).

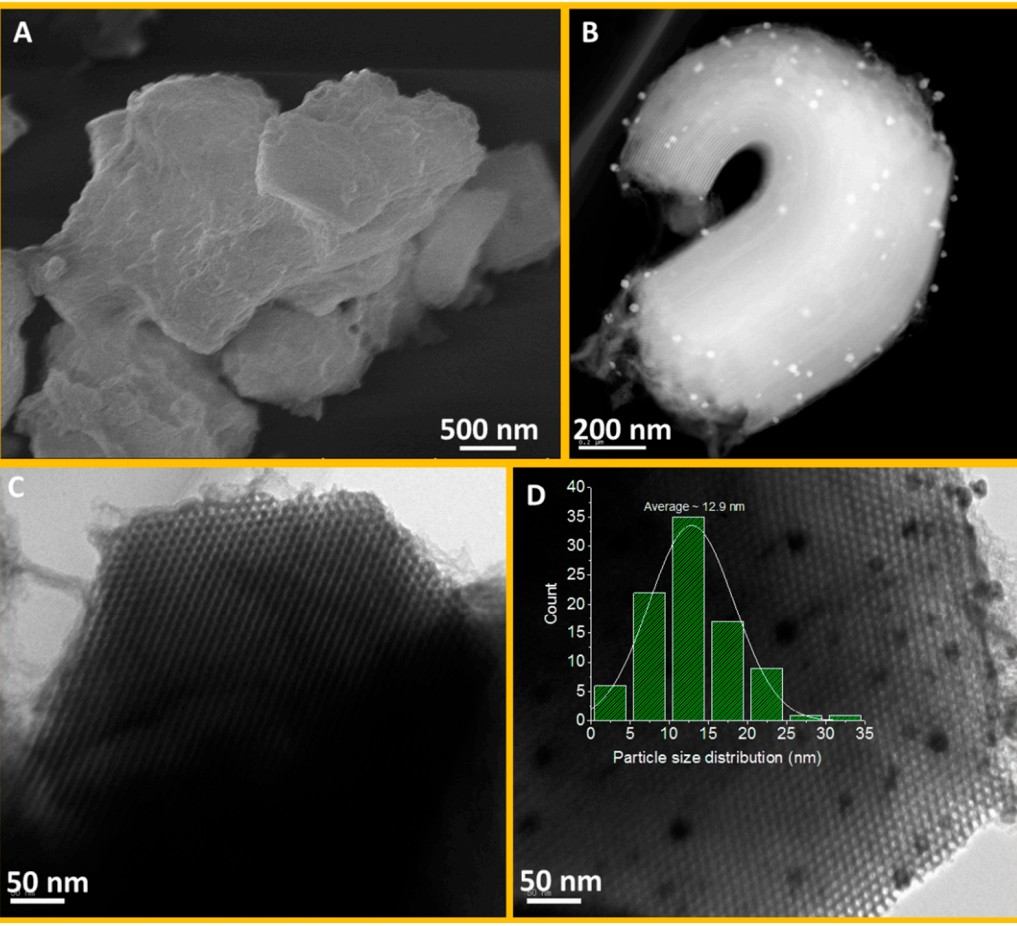

**Figure 4.** (**A**) SEM and (**B**) HAADF-STEM images of Pd@MO-ISO. TEM images of (**C**) MO-ISO and (**D**) Pd@MO-ISO. Inset indicates the pore size distribution of Pd nanoparticles measured by HAADF-STEM and TEM images.

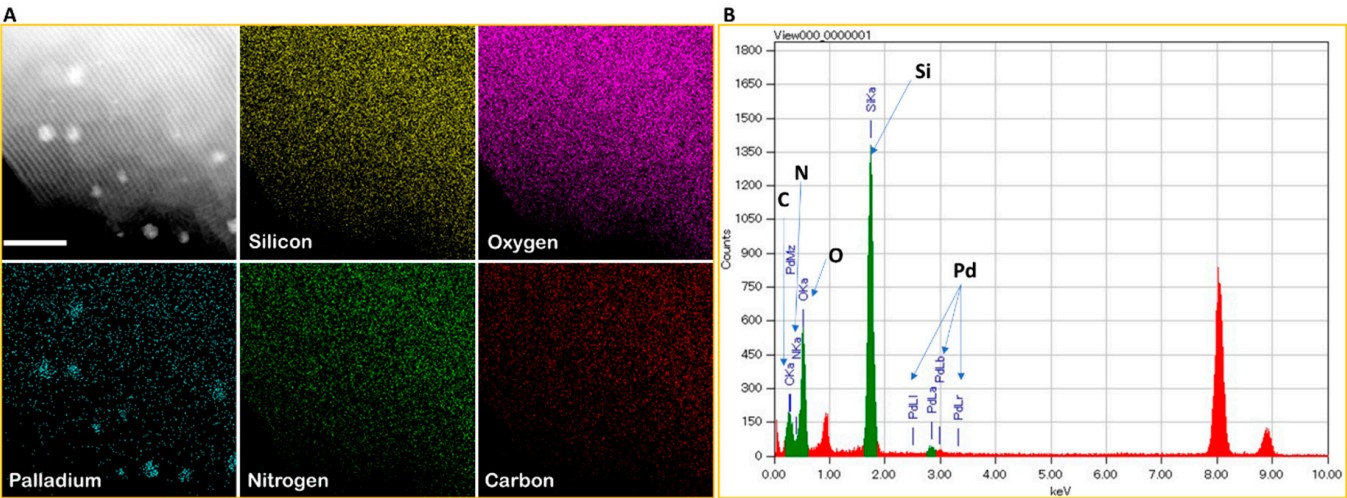

**Figure 5.** (**A**) HAADF-STEM image and elemental TEM-mapping images of every element in a separate image related to Pd@MO-ISO. Scale bar shows 100 nm and is constant for all images. (**B**) TEM-EDS spectra of Pd@MO-ISO.

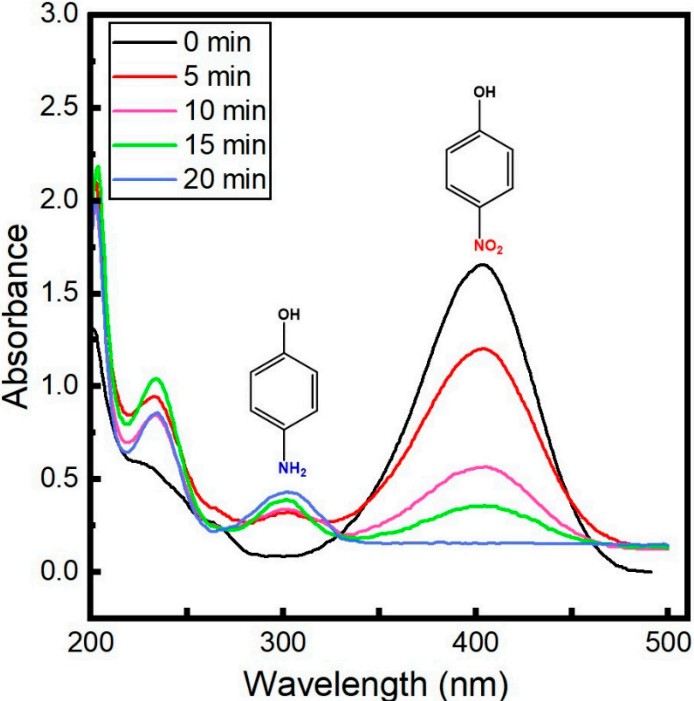

**Figure 6.** Scanning the UV–Visible spectra of PNP solution that catalytically reduces to PAP within 20 min.

We also studied the recyclability of the catalyst, Pd@MO-ISO, as the recyclability is one of the pivotal features in heterogeneous catalysts in organic transformation. Accordingly, we used the optimised reaction conditions, i.e., 3 mM PNP, room temperature, aqueous conditions, and 5 mmol $NaBH_4$. The results show that Pd@MO-ISO as a heterogeneous catalyst that is easily recoverable and reusable at least for five consecutive cycles (Figure 7A). In addition, the analysis of the reaction solution, once the catalyst is filtered, through atomic absorption spectroscopy (AAS) in the fifth cycle, confirmed that the Pd leaching is negligible (<1%). SEM-based EDS spectra of recovered Pd@MO-ISO after five cycles also shows the presence of Pd species, further confirming the resistance of the material in Pd leaching (Figure 7B). However, the observed loss, could be attributed to the relative collapse of the silica mesoscopic structure by exposing to the highly reactive $NaBH_4$.

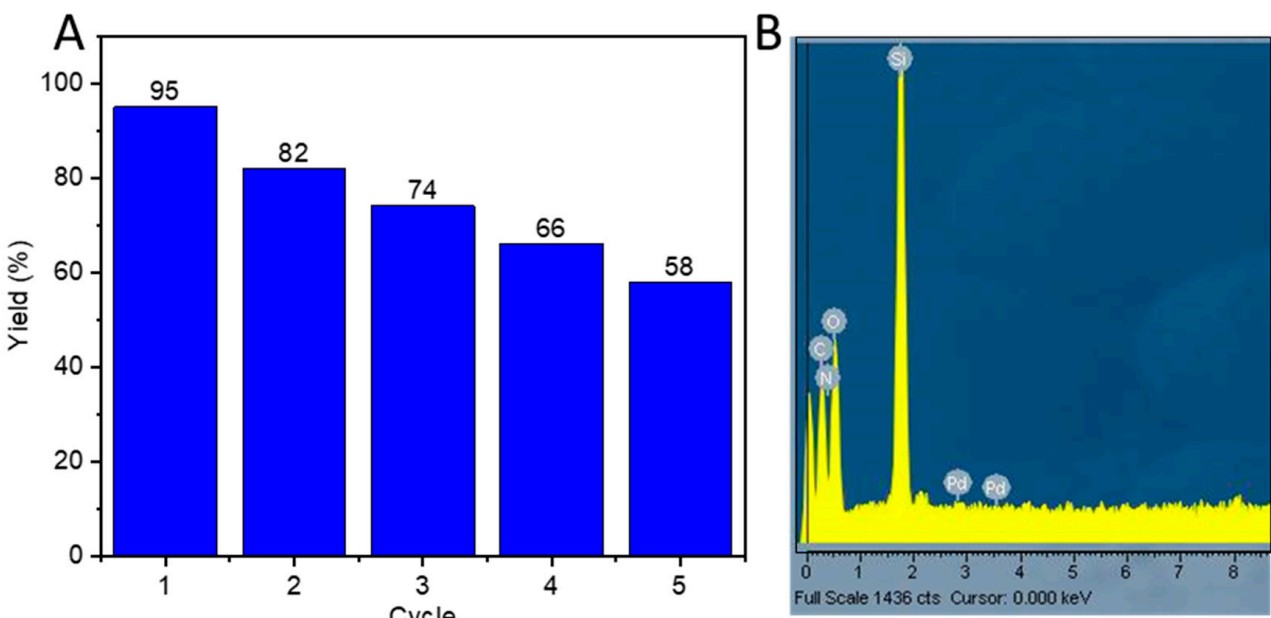

**Figure 7.** (**A**) Recyclability of the Pd@MO-ISO as catalyst in the chemical reduction of PNP into PAP in five consecutive cycles. (**B**) EDS spectra of PD@MO-ISO.

We further compared the catalytic results of our catalyst and method with the previously reported catalysts, which are used for PNP reduction to PAP. We found our catalyst superior to those of several catalysts and methods reported before, in terms of TON (Table 1).

**Table 1.** Comparing the PNP reduction through our synthesised catalyst with the previously reported catalysts in aqueous solution in the presence of $NaBH_4$ as reducing agent.

| Catalyst | Time (min) | Temperature (°C) | TON | Reference |
|---|---|---|---|---|
| TA@Fe$_3$O$_4$-AgNPs | 6 | r.t. * | 25 | [44] |
| PdPt@Chitosan | 120 | 22 | 10 | [45] |
| Pd@MO-urea | 26 | r.t. | 64 | [17] |
| Pd@MO-ISO | 20 | r.t. | 125 | This work |

* r.t. is room temperature.

## 3. Materials and Methods

### 3.1. Materials Characterisations

The SEM images were observed with a HITACHI SU-8230 scanning SEM. TEM images were taken with a JEOL JEM-2100F microscope (operated at 300 kV). $N_2$ adsorption–desorption, BJH, and BET analyses were carried out at 77 K using a Microtrac Bel BEL-mini. Prior to the measurements, the samples were evacuated at 90 °C for 20–24 h. ICP-OES was performed by Pekin-Elmer (Waltham, MA, USA) model. The pore size distribution of the mesopores were determined by BJH analysis obtained by Autosorb Microtrac Bel BEL-mini.

### 3.2. Materials Synthesis

For the synthesis of this organosiloxane bridge, THEIC (3 mmol, 0.783 g) was reacted with (3-isocyanatopropyl) triethoxysilane (1 mmol, 0.5 mL) at 135 °C for 3 h, cooled to 80 °C, and stirred at that temperature for another 3 h. Then, a colourless oily product was obtained at the end, which was used without further purifications [16]. The synthesised isocyanurate-based organosiloxane bridge was further employed in the synthesis of the corresponding MO through co-condensation of TEOS. Accordingly, the Pluronic triblock copolymer P123 (2 g, MW = 5800 g·mol$^{-1}$) was dissolved in HCl aqueous solution ($10^{-4}$ M,

75 mL) and stirred for 3 h. Then, the as-synthesised organosiloxane bridge from the previous step was mixed with TEOS with the molar ratio of 1:15, respectively, and co-condensed in the P123/HCl aqueous solution mixture and stirred for 24 h at 25 °C. After stirring for 24 h, the reaction mixture was transferred to an autoclave for aging for 24 h at 100 °C. After aging, the polymeric P123 was extracted from the pore space by Soxhlet with MeOH for 4 days. A white powder as the final product after drying at 60 °C for 4 h in an oven was obtained. For simplicity, the product obtained in this stage was named "MO-ISO."

$Na_2PdCl_4$ (37 μmol, 0.011 g) in acetonitrile was added dropwise to MO-ISO (0.2 g). Afterward, creamy pasty product was collected and dried at 100 °C. After drying for 3 h, it was re-dispersed in MeOH and then, $NaBH_4$ was added to the dispersion and allowed to stir for 0.5 h. Finally, the final product was centrifuged and washed for three times with EtOH (10 mL) and dried in oven at 60 °C for 3 h. For simplicity, the product obtained in this step was named "Pd@MO-ISO." The wt% of Pd loaded in MO-ISO was analysed by ICP-OES technique, indicating that the Pd content is 2 wt% in the MO-ISO.

*3.3. Catalytic Test*

In the catalytic test to reduce PNP to PAP, Pd@MO-ISO (5 mg) was dispersed in the aqueous PNP solution (3 mM, 40 mL) by sonicating for 5 min. Then, the sodium borohydride (5 mmol, 125 mg) was added to the solution and continued to stir vigorously. During the reaction, every 5 min, the sampling from reaction progress was achieved by a syringe equipped with syringe filter to separate the catalyst from the reaction media. Then, the obtained samples from each minute were analysed by UV–VIS spectrometer to monitor the reaction progress.

## 4. Conclusions

Here, we presented a heterogeneous green catalyst on the basis of the new Pd-based hybrid mesoporous organosilica, which could successfully adsorb Pd ions and support it on the surface. The Pd supporting process had not a significant destructive effect on the mesoscopic structure of the MO-ISO and had an efficient catalytic activity in the chemical reduction of PNP to PAP via $NaBH_4$ as reducing agent. The catalyst also showed a high rate of recyclability and negligible Pd species leaching over recycling the catalyst. The catalytic activity of Pd@MO-ISO was excellent in comparison to other previously reported catalysts with the similar textural structure.

**Author Contributions:** Conceptualization, E.D. and M.G.D.; methodology, A.Z.; software, A.Z.; validation, E.D., A.Z.; formal analysis, A.Z.; investigation, A.Z.; resources, M.G.D.; data curation, A.Z.; writing—original draft preparation, A.Z.; writing—review and editing, E.D. and M.G.D.; visualization, A.Z.; supervision, M.G.D. and E.D.; project administration, M.G.D. and E.D. All authors have read and agreed to the published version of the manuscript.

**Funding:** This research received no external funding.

**Conflicts of Interest:** The authors declare no conflict of interest.

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
