# Peer review of "The Isocyanurate-Carbamate-Bridged Hybrid Mesoporous Organosilica: An Exceptional Anchor for Pd Nanoparticles and a Unique Catalyst for Nitroaromatics Reduction"

_catalysts, doi:10.3390/catal11050621_

Round 1

Reviewer 1 Report

All my concerns regarding the previous version of this manuscript were correctly addressed. I recommend publication after some additional corrections:

  • "isocyanurate" in the title
  • lines 41-42: "it is an overwhelming process usually deals with pore-blocking..." I don't understand this sentence. Please rewrite it.
  • in part 2.1: I would add a sentence or a paragraph on the deposition of the palladium nanoparticles and the quantity of palladium deposited on the MO-ISO. Currently, the preparation and characterization of the support are described but not the incorporation of palladium, which is surprising. The reader is informed of the impact of the palladium on the surface area of the surface (line 79) but at this stage, doesn't even know how the palladium was incorporated (and how much).
  • line 96: "two peaks in the 13C-NMR spectrum
  • line 128: TON is Turn Over Number not frequency
  • line 134: "We examined the reduction reaction of versus the time" I don't understand the sentence, please rewrite it.
  • lines 145-146 and figure 7: in response to my remark, the authors explained that they suspected that the decrease in catalytic activity is due to the degradation of the mesoporosity of the support in the reaction conditions. I think that it is a piece of very important information that should figure in the manuscript. Please add a sentence explaining this to the readers with some bibliographical references (if some previous articles already noted this kind of degradation).
  • Table 1: the two columns "reductant" and "solvent" are strictly identical for all the catalysts presented. I would delete these two columns and add the information on the solvent and the nature of the reductant in the legend of the table. 
  • in the experimental part (line 180): which quantity of MO-ISO was dispersed in acetonitrile?
  • line 187: please add some details on the analysis by ICP-OES. For example, how was the sample from Pd@MO-ISO prepared for this analysis?

Author Response

Many thanks for reviewing our paper and giving constructive comments toward our manuscript. We have addressed and applied all comments one by one and have attached the response letter below for your reference. 

Reviewer 2 Report

Manuscript ID: catalysts-1219257

Title: The isocynurate-carbamate bridged hybrid mesoporous organosilica; an exceptional anchor for Pd nanoparticles and a unique catalyst for nitroaromatics reduction

Authors: Ali Zebardasti, Mohammad G. Dekamin*, Esmail Doustkhah*

This manuscript presents the catalytic application of Pd nanoparticles-supporting mesoporous organosilica in the reduction of 4-nitrophenol. The authors performed N2 adsorption/ATR-FTIR/SEM/TEM/STEM-EDS/UV-Vis/AAS analyses for detailed characterization of the silica support and Pd nanoparticles. It seemed that the authors have fully addressed previously raised concerns by three reviewers. In this respect, I recommend this manuscript for consideration of acceptance (or publication). For further improvement, I also ask some questions on this work and hope to give several suggestions as written below: The notations (e.g., RA, RC, PHC, or CoR) are not easy to know the meaning or full name of the method.

  1. The material itself, except for the loading of Pd nanoparticles, was previously reported by the authors [Inorg. Chem. 2020, 59, 11223-11227, Ref. 16.]. How about highlighting more on the catalytic aspect and metal loading? The current title looked like a report on the synthesis method of new silica support as well as the catalytic application of supported Pd nanocatalyst.
  2. Please cite original papers about SBA-15, such as Science 279 (5350), 548-552 & Chem. Mater. 2000, 12, 1961-1968.
  3. Also cite recent papers about Pd supported organosilica materials if necessary, such as Catalysts 2019, 9, 140, MMM 2010, 131 (1–3), 162-169. Then, assess the exceptional advantage of the isocynurate-carbamate bridged hybrid mesoporous organosilica, compared with other organosilica materials in the papers and Ref. 20. In other words, what is the exact role of isocynurate-carbamate moiety for enhancing the catalytic activity of supported Pd nanoparticles, as compared to other silica-supported Pd catalysts?
  4. In Figure 2, the x-axis values of Figure 2a are somewhat awkward. Please change them to 0.2, 0.4, and so on. Is there any offset in the data set? Please specify the offset amount in the figure caption or other possible area. The authors may think of the pure mass of silica itself rather than the mass of the Pd-silica composite. The y axis of Figure 2b should be dV/dD or dV/dlogD in general. What is the method exactly for estimating the pore size distribution? Clearly state the method in the experimental section or in the caption, at least cite the previous paper in the relevant position.
  5. The reasoning for the decreasing catalytic activity is not convincing. There was a 14% loss of efficiency after 1st cycle and 39% after 4th cycle. This value is not negligible, although the leached amount of Pd was negligible. However, the relevant explanation in the revised (or submitted) manuscript did not explain the reason for the catalyst deactivation and possible regeneration methods. Was there any other by-product(s) during the reaction? Does the yield become ~100% if the reaction were continued for a longer time than 20 min?
  6. The authors used only one Pd loading amount (i.e., 2 wt%) for the use as catalyst sample. Was it the optimized loading amount for the best catalytic performance? They should have shown or discussed the optimization process at least in supporting information. Is there any reason for choosing the Pd precursor for this application?
  7. Please provide more information on the size of Pd nanoparticles. Mean diameter should be mentioned in the manuscript after counted by (S)TEM images, or other possible adsorption-based methods. This is very important to evaluate the TON, since the number of the active site is determined by the surface of Pd particles rather than total Pd atoms. Regarding this, please write the definition of the active site used for TON calculation in this work.
  8. Abbreviations should be placed in the very first part, not in the experimental section.

Author Response

(The authors gave the same response as above.)

Round 2

Reviewer 2 Report

The revised version of this manuscript and relevant replies well addressed the raised concerns from the two reviewers. I would like to recommend it for publication.